Land-use change influence soil quality parameters at an ecologically fragile area of YongDeng County of Gansu Province, China

Adingo Samuel 1
Yu Jie-Ru 2
Xuelu Liu liuxl@gsau.edu.cn 2
Jing Sun 2
Li Xiaodan 3
Xiaoning Zhang 1
1 College of Forestry, Gansu Agricultural University , Lanzhou , Gansu Province , China
2 College of Resources and Environment, Gansu Agricultural University , Lanzhou , Gansu Province , China
3 School of Management, Gansu Agricultural University , Lanzhou , Gansu Province , China
Li Chenxi
Electronic publication date: 2021 Oct 27
Publication date: 2021
Volume: 9
Electronic Location ID: e12246
Received 2021 Jul 27; Accepted 2021 Sep 13
Copyright: ©2021 Adingo et al.
Copyright year: 2021
Copyright holder: Adingo et al.
License: This is an open access article distributed under the terms of the Creative Commons Attribution License, which permits unrestricted use, distribution, reproduction and adaptation in any medium and for any purpose provided that it is properly attributed. For attribution, the original author(s), title, publication source (PeerJ) and either DOI or URL of the article must be cited.
License URL: https://creativecommons.org/licenses/by/4.0/

Keywords: Land-use change, Principal component analysis, Soil quality, Soil fertility, Physicochemical properties

Funding: “Research on the Coordination Relationship between Land Urbanization and Population Urbanization” (project No. GSAU-ZL-2015-046) Fundamental Research Funds of Gansu Provincial Natural Science Fund of “Research on Land use Ecological Security in Ecologically Vulnerable Areas” (project No. GSAN-ZL-2015-045) This work was supported by “Research on the Coordination Relationship between Land Urbanization and Population Urbanization” (project No. GSAU-ZL-2015-046) and Fundamental Research Funds of Gansu Provincial Natural Science Fund of “Research on Land use and Ecological Security in Ecologically Vulnerable Areas” (project No. GSAN-ZL-2015-045). The funders had no role in study design, data collection and analysis, decision to publish, or preparation of the manuscript.

==============================
Dry ecosystems, despite their relative levels of aridity, are very diverse, and play a vital role in the livelihoods of many dryland inhabitants. It is therefore critical to investigate the relationship between land-use change and soil quality parameters to offer a scientific basis for optimizing land-use planning and improving soil quality status in dry ecosystems and ecologically vulnerable areas. This study, therefore, analyzed the physicochemical properties of soils in five different land-use types namely farmland, abandoned farmland, natural grassland, artificial lemon forest, and poplar woodland at YongDeng County. The soil quality status of the aforementioned land-use types was also evaluated through Principal component analysis. The results revealed that abandoned farmland and natural grassland recorded the highest average values of soil coarse particles of 24.0% and 23.4% respectively compared to the other land-use types. The highest average value (46.1%) of fine soil particles was recorded in poplar woodland followed by natural grassland (36.6%) and the average value of very fine soil particles was higher in farmland (40.8%) and artificial lemon woodland (38.3%) than in the other land-use types. The average value of clayey particles was highest in farmland (11.1%), followed by artificial lemon woodland (9.3%), and abandoned farmland (6.5%), then poplar woodland which recorded an average value of (4.2%). The average values of Soil water content, soil pH, soil electrical conductivity, and soil total nitrogen content were significantly higher in farmland compared to the other land-use types. Soil organic carbon content was significantly higher in abandoned farmland at (P < 0.03) and lemon woodland at (P < 0.01) than in farmlands, natural grasslands, and poplar stands. The soil quality indicators of the different land-use types were significantly correlated with each other. Among them, the correlation coefficient of each evaluation index was highest in poplar woodland, followed by natural grassland, lower in farmland and artificial lemon woodland, and lowest in abandoned farmland. The overall soil quality scores were in the following order: farmland > abandoned farmland > 0 > grassland > lemon woodland > poplar woodland. In the study area, the soil quality of farmland that has been finely managed and naturally restored to grassland following abandonment is superior, whereas the soil quality of natural grassland, artificial lemon woodland, and poplar forest land is substandard. The comprehensive analysis of soil quality demonstrates that conservation tillage and fine management of water-irrigated farmland, as well as the natural conversion of abandoned farmland to grassland, can significantly improve the soil quality of sandy soils, reduce water and soil loss, increase fertility, and gradually improve regional ecological environmental conditions.

Introduction

Soil is a balance of physical, chemical, and biological processes in nature, and is also an independent historical natural body (Manpoong & Tripathi, 2019). The quality of soil is directly or indirectly related to the food quality of people (and animals), the quality of the living environment, the health status, and the well-being of the breeding offspring (Jobbágy & Jackson, 2000). Over the past years, there have been concerted efforts by relevant stakeholders to achieve the goal of sustainable agriculture to feed the ever-growing population of the world, which is estimated to reach 9 billion by the year 2050 (Kyte, 2014; Lammerts van Eekeren et al., 2018; Yang et al., 2019; Sahu et al., 2020). However, attaining sustainable agriculture remains a mirage as it has been greeted with challenges such as climate change (Calicioglu et al., 2019) which has resulted in unreliable and irregular rainfall patterns, pests and diseases, and loss of important soil resources due to soil erosion (Ding et al., 2013).

The aim of land use, land management, climatic conditions, and intrinsic soil qualities all influence changes in soil parameters (Zhang et al., 2018). Soil deterioration can be caused by the use of land for cultivation or development, as well as the removal of natural vegetation and disruption of the soil profile (Shekhovtseva & Mal’tseva, 2015). Converting native vegetation to farmland lowers soil nutrient and the content of soil organic carbon and at the same time, increase bulk density (Zucca, Canu & Previtali, 2010) and sand content (Biro et al., 2013). In addition, the genetic soil layer is highly altered in urban areas, resulting in soil constriction and loss of soil organic carbon (SOC) (Zambon et al., 2018). Urban soils, on the other hand, are regarded as carbon sinks (Weissert, Salmond & Schwendenmann, 2016). Long-term irrigation, for example, can enhance soil carbon and nitrogen content while also improving soil quality (Fiorini et al., 2020). Agricultural wastes can also help with carbon and nitrogen sequestration in the soil (Deng et al., 2016).

Nevertheless, in arid areas, changes in soil organic carbon and nutrient content resulting from cropland abandonment are largely determined by soil and climatic conditions (Kosmas, Gerontidis & Marathianou, 2000). Evans & Belnap (1999) showed that arid ecosystems experience a loss of soil nutrients due to land-use change. As a result, soil quality deteriorates, and soils become ecologically vulnerable, leading to soil degradation (Lal, 2015) and the weakening of erosion resistance (Khalili Moghadam et al., 2015). Jeong & Dorn (2019) observed that in an area with a desert climate, land-use changes associated with urbanization accelerate soil erosion. Land-use changes (e.g., from a forest or natural grassland to cropland and pasture) on soil carbon and other nutrients is well documented (Smith et al., 2016). Moreover, studies have mostly focused on the characteristics of urban pollution following land-use change and associated soil properties such as SOC, pH, and soil N (Yang et al., 2015). However, cities are considered to be highly disturbed ecosystems and in arid regions, the response of soil properties to these extreme disturbances is still unknown. To understand the dramatic changes in land use and their impacts on ecosystems, further research is needed to explore the extent to which changes in land use affect soil properties in arid environments. A better understanding of these impacts is essential to optimize land use and soil conservation and minimize land degradation, especially in fragile ecosystems.

YongDeng County in China’s Gansu Province has experienced significant expansion, progress, and development in areas such as agriculture and construction. The county’s growth, production, and expansion in various industries have led to a rapid increase in land use. Due to its special ecological characteristics, natural geographic location, and recent land degradation, the state has taken measures such as returning farmland to forest and grass and grassland sealing to improve the ecological environment, reduce the degree of desertification, and facilitate the recovery of vegetation. At present, there are mainly five types of land use in the area: farmland, abandoned farmland, natural grassland, lemon woodland, and poplar woodland. In this study, we seek to investigate the distribution of surface vegetation and determine the physical and chemical characteristics of soil in the different land-use types in the area and also use principal component analysis to comprehensively evaluate the soil quality status of the aforementioned land-use types, aiming to elucidate the influence of land-use change on soil quality and to provide a basis for “returning farmland to forests and grasslands” and adopting reasonable land-use policies in the area.

Materials & Methods

Research area overview

The location of the research area is depicted in Fig. 1. It is located between the latitudes of 102°36′ and 103°45′ East and 36°12′ and 37°07′ North. It has a typical continental monsoon climate of the mid-temperate range. Annual rainfall is estimated to range between 30 and 600 mm, with a steadily decreasing southeast-to-northwest gradient, with summer accounting for 58.8 percent of total precipitation, and significant inter-annual variability of precipitation. Evapotranspiration is 2,710 mm per year, and the frost-free period is 120 days per year. The annual average temperature is 7.7 °C, and the climate varies significantly between winter and summer, with an average temperature difference of 28 °C. The average annual wind speed is 2.8 ms−1, and winter and spring bring plenty of sandy weather, with 323 sand rise greater than 5 ms−1 per year.

Figure 1 Map of the study area and sampling points.

Map of the study area and sampling points.

The topography of the area is naturally hilly, with stony mountains and loess hills. The cliffs are interconnected, while the plains are undulating, with the Yellow River running through them. Within YongDeng County’s elevation range of 1,000–3,000 m are the Loess highlands, Inner Mongolia, and the Qinghai-Tibet plateau. The region’s zoned soils are primarily yellow cotton soil, gray calcium soil, and light gray calcium soil; the region’s non-zoned soils are primarily sandy soil, saline soil, and meadow soil, among others, with sandy soil being particularly prevalent in the region’s north-central part. The soil texture is a predominantly light loam, sandy loam, and sandy soil, all of which have a loose structure and low fertility. The soil type used in this study is pedocals (calcareous) which are neutral to alkaline in reaction.

The study site included five distinct land uses: farmland (F), abandoned farmland (Q), natural grassland (G), artificial lemon woodland (N), and poplar woodland (Y). Maize (Zea mays) is the predominant crop planted by farmers in this study area, with an average height of 2.52 m, row spacing of 75 cm, plant spacing of 25 cm, and an area of 11.8 hm2. Simultaneously, artificially planted lemon and poplar trees are frequently used to promote vegetation restoration in severely degraded areas, preventing wind erosion and aiding in sand fixation, and enhancing the soil environment. The average canopy width of an artificial lemon tree is 0.42 m2, the average plant density is 0.52 plants per m2, and covers an area of 15.3 hm2, with plants spaced 7 m apart and rows spaced between 5 and 8 m apart. The average height of an artificially planted poplar tree is 4.5 m, with a diameter of 9 cm at breast height, a row spacing of 8–10 m, and a surface area of 14.0 hm2. Natural grassland in the study area occupies approximately 13.3 hectares. Artemisia scoparia, Pennisetum centrasiaticum, and Lespedeza potaninii dominate the surface vegetation in this area as shown in (Table 1).

Table 1 The basic survey of land use types.

Sample Land	Number of surface vegetation species/species	Surface vegetation average density/ (stand m−2)	Surface vegetation average height/cm	
F	–	–	–	
Q	3.196 ± 0.008b	71.490 ± 0.350b	8.420 ± 0.005b	
G	4.396 ± 0.003a	54.546 ± 0.526d	8.626 ± 0.008b	
N	3.203 ± .008b	60.426 ± 0.396c	9.573 ± 0.331a	
Y	2.400 ± 0.214c	82.080 ± 3.188a	7.170 ± 0.006c	
Notes.

F farmland

Q abandoned farmland

G natural grassland

N artificial lemon woodland

Y poplar woodland

Lowercase letters indicate significant differences between land-use types at (P < 0.05).

Experimental design

In the summer of 2020 (end of July), five different land-use types were chosen as study plots in the study area: farmland (maize field), abandoned farmland, natural grassland, artificial lemon woodland land, and poplar woodland land. Within each study plot, four (4) soil sampling sites were located at the two diagonal corners and the center of the plot. Soils at each sampling site were collected from soil depth of 0–10 cm using a 5 cm diameter soil auger with three replicates each, which were homogenized by hand mixing thoroughly to obtain composite sample (each sample was not less than 1 kg). Samples were placed in well-labeled plastic bags and transported immediately to the laboratory. A total of sixty (60) soil samples (5 types of sample plots × 4 sampling points × 3 replicate samples) were collected in all the study plots. In abandoned farmland, natural grassland, artificial lemon woodland, and poplar woodland, one herbaceous survey sampling Quadrat with an area of 1  × 1 m2 was set up and random sampling method used to survey the surface herbaceous vegetation, to determine the number of individuals (plant m−2), abundance, and height (cm) as the background of the sample plots (Table 1). This was repeated three times in the respective study plot with an interval of more than 15 m between sampled points.

Soil sample collection and soil parameters determination

Surface mixed soil samples (0–10 cm) were taken at each survey sample site for soil index determination. First, 1/4 of the fresh soil samples collected were used for soil moisture content determination. Then, the remaining 3/4 of the soil samples were passed through a two mm soil sieve to remove impurities such as plant roots and foreign material such as dead plant parts, old manure, gravels, and compost to reduce variation which may occur as a result of dilution of organic matter due to mixing through cultivation and other factors. Samples were then air-dried in a natural state and then used to determine soil particle size composition, pH, electrical conductivity, organic carbon and total nitrogen content.

Soil moisture content (%) was determined by the drying method, in which an aluminum box containing fresh soil samples was weighed on an analytical balance to the nearest 0.01 g. The samples were then placed in an oven at 105 °C for 24 h and weighed immediately after cooling to room temperature. Soil particle size composition was determined using a Mastersizer 3000 laser diffraction particle size analyzer (Malvern, Uk-LD_3000) using an array of 52 detectors with a repeatability error ≤ ±0.5%, and an accuracy error ≤ ±1%. Soil texture was classified according to the United States Department of Agriculture (USDA) soil texture classification criteria: coarse soil particles (250–1,000 µm), fine soil particles (100–250 µm), very fine soil particles (50–100 µm), clay particles (<50 µm) (Shangguan et al., 2014). Soil pH value (water-soil ratio suspension ratio of 2.5:1), soil electrical conductivity (µs m−1) (5:1 for water-soil ratio leachate) were determined with a P4 multifunctional measuring instrument. Soil organic carbon (g kg−1) was determined by the potassium dichromate external heating method, and soil total nitrogen (g kg−1) was determined by the semi-micro Kjeldahl method (Sáez-Plaza et al., 2013).

Soil quality evaluation method

Selection of evaluation indicators

This paper selected and established a comprehensive evaluation index system appropriate for this study based on the research on soil quality evaluation systems by Cao et al. (2021). Nine distinct indicators were selected including coarse soil particles (X1), fine soil particles (X2), very fine soil particles (X3), clay soil particles (X4), soil water content (X5), soil pH value (X6), soil electrical conductivity (X7), soil organic carbon (X8), and soil total nitrogen (X9).

Principal component analysis steps followed in this work are as follows: (1) the original data matrix X, which includes the aforementioned nine soil indicators were enumerated. (2) Standardization of raw data. Because each index has a different magnitude, the raw data for each measured index was standardized to eliminate the effect of different magnitude on the evaluation results (Vidal, Ma & Sastry, 2016). The standardization formula is as follows; (1) Xij=Xij−XjSj

where Xij is the standardized value of the j th indicator of the i-th sample plot; Xij is the measured value of the j th indicator of the i-th sample plot; Xij isthe average value of the j th index; Sj is the standard deviation of the j th indicator; i is the number of sample plots (i = 1, 2, 3, 4, 5 in this paper); j is the number of indicators selected in the evaluation (j = 1, 2, ….9 in this article). (3) The sample’s correlation matrix R was calculated. (4) Jacobi’s method was used to evaluate the eigenvalues and eigenvectors of the correlation matrix R (Demmel & Veselić, 1992). (5) The contribution rate and cumulative contribution rate, as well as the number of principal components, their meanings, were determined and principal component equations established. (6) The factor loading of each principal component and the indicator’s common factor variance were calculated. Each indicator’s weight wj was calculated using the equation below; (2) ∑j−1nwj=1

(7) The comprehensive evaluation value of soil quality for each land-use type was calculated based on the weight of each index, and the soil quality of different land-use types was evaluated accordingly. The comprehensive evaluation values of soil quality for different land-use types were determined by the equation; (3) Pi= ∑j−1nXijwj

where Pi is the comprehensive evaluation value of soil quality of the i-th land-use type; Xij is the standardized value of the jth indicator of the i-th land-use type; wj is the weight value of the j th indicator, obtained by principal component analysis; i is the number of sample sites (i = 1, 2, 3, 4, 5 in this paper); n is the number of indicators selected in the evaluation (n = 9 in this paper).

Data processing and analysis

All data were statistically analyzed using SPSS20.0 software. One-way analysis of variance (One-Way ANOVA) and least significant differences (LSD) method was used to analyze the differences between different data groups, and Spearman’s correlation coefficient was used to analyze the correlation between different indicators. The significance level was (P = 0.05).

Results

Impact of land use change on Soil physical and chemical properties

Soil particle size composition

As shown in (Table 2), there were significant differences (P < 0.05) in soil particle composition among different land-use types. Among them, abandoned farmland and natural grassland, recorded the highest average values of soil coarse particles of 24% and 23% respectively and were significantly higher than farmland, artificial lemon woodland, and poplar woodland which recorded average values of 18%, 18% and 16% respectively, but there was no significant difference (P > 0.05) between farmland and artificial lemon woodland. The highest average value (46%) of fine soil particles was recorded in poplar woodland followed by natural grassland (3%) whiles farmland, abandoned farmland and artificial lemon woodland recorded fine soil particles content values of 28%, 35% and 33% respectively. The average value of very fine soil particles was higher in farmland (40%) and artificial lemon woodland (38%) than in abandoned farmland (33%), natural grassland (32%), and poplar woodland (32%). The average value of clayey particles was significantly higher in farmland (11%), artificial lemon woodland (9%), and abandoned farmland (6%), than poplar woodland, which recorded an average value of (4%).

Table 2 Variation of soil particle composition under land use types (%).

Land use type	Coarse soil particles	Fine soil particles	Very fine soil particles	Clay soil particle	
F	18.860 ± 0.082b	28.870 ± 0.078d	40.893 ± 0.133a	11.130 ± 0.236a	
Q	24.016 ± 0.170a	35.703 ± 0.574b	33.133 ± 0.240c	6.526 ± 0.095c	
G	23.403 ± 1.171a	36.643 ± 1.124b	32.346 ± 0.574c	6.013 ± 0.006cd	
N	18.053 ± 0.114b	33.386 ± 0.916c	38.313 ± 0.288b	9.323 ± 0.03512b	
Y	16.156 ± 0.238c	46.143 ± 0.323a	32.713 ± 0.540c	4.246 ± 0.035e	
Notes.

F farmland

Q abandoned farmland

G natural grassland

N artificial lemon woodland

Y poplar woodland

Lowercase letters indicate significant differences between land-use types at (P < 0.05).

Soil moisture

As illustrated in (Fig. 2), there was a significant difference (P < 0.05) between the soil moisture contents among the land-use types. The highest average value (4%) of soil water content was found in farmland, compared to abandoned farmland, natural grassland, artificial lemon woodland and poplar woodland, which recorded average soil water content values of 1%, 0.9%, 0.7% and 0.8% respectively. The soil water content of abandoned farmland was significantly higher than that of artificial lemon woodland and poplar forest land (P < 0.03), while natural grassland was in between. There were no significant differences (P > 0.06) between natural grassland and the abandoned farmland, and between natural grassland and artificial lemon woodland and poplar woodland.

Figure 2 Variation of soil moisture content under different land use types.

Soil pH and electrical conductivity

From (Fig. 3), it can be seen that the soils in this area are weakly alkaline, and there were significant differences (P < 0.05) among different land-use types. The soil pH value was highest in farmland (8) and was significantly higher than abandoned farmland, natural grassland, artificial lemon woodland, and poplar woodland (P <  0.03), and abandoned farmland and natural grassland recorded soil pH values of 7 and 7 respectively which were significantly higher than artificial lemon woodland and poplar woodland (P <  0.05) but there was no significant difference between abandoned farmland and grassland and between artificial lemon woodland and poplar forest land (P > 0.05). As shown in (Fig. 4), the soil electrical conductivity of farmland was significantly higher than that of abandoned farmland, natural grassland, artificial lemon woodland, and poplar woodland (P < 0.04), but there was no significant difference between the latter four (P > 0.07).

Figure 3 Variation of soil pH under land use types.

Figure 4 Variation of electrical conductivity under different land use types.

Soil organic carbon and total nitrogen

As shown in (Fig. 5), there was a significant difference (P < 0.05) in soil organic carbon among different land-use types. The highest soil organic carbon (5 g kg−1) was found in abandoned farmland which was significantly higher at (P < 0.04) than that of farmland, natural grassland, lemon woodland, and poplar woodland which recorded soil organic carbon values of 4 g kg−1, 3 g kg−1, 4 g kg−1 and 2 g kg−1 respectively. There was no significant difference in the soil organic carbon content of farmland, and lemon woodland (P > 0.07) but they were significantly higher than the poplar woodland (P < 0.02).

Figure 5 Variation of soil organic carbon under different land use types.

As can be seen from (Fig. 6), the total N of the soil was significantly higher (P < 0.04) in farmland which recorded a total N average value of 0.6 g kg−1 than in abandoned farmland, natural grassland, lemon woodland, and poplar woodland which recorded average values of 0.5 g kg−1, 0.3 g kg−1, 0.4 g kg−1 and 0.2 g kg−1 respectively. The average total N value of abandoned farmland was significantly higher (P <  0.03) than that of natural grassland, lemon woodland, and poplar woodland. The average value of total N of artificial lemon woodland was significantly higher than that of poplar forest land (P < 0.04), but there was no significant difference between artificial lemon woodland and natural grassland (P > 0.06).

Figure 6 Variation of total nitrogen under different land use types.

Correlation analysis between soil quality indicators of different land-use types

As can be seen from (Table 3), in farmland, coarse soil particles were negatively correlated (−0.991) with very fine soil particles; fine soil particles were negatively correlated (−0.901) with clay soil particles; very fine soil particles were negatively correlated (−0.905) with soil water content; soil water content was positively correlated (0.913) with soil total nitrogen. There was no correlation among all indicators in abandoned farmland. In natural grassland, coarse soil particles were negatively correlated (−0.893) with very fine soil particles, and positively correlated (0.927*) with soil organic carbon; soil water content was negatively correlated (−0.902) with soil electrical conductivity and soil total nitrogen (−0.993); Soil electrical conductivity was positively correlated (0.920) with total soil nitrogen. In artificial lemon woodland, coarse soil particles were negatively correlated (−0.912) with fine soil particles and very fine soil particles (−0.94); soil water content was positively correlated (0.895) with soil electrical conductivity. In poplar woodland, coarse soil particles were negatively correlated with very fine soil particles (−0.942), soil total nitrogen (−0.942), and soil organic carbon (−0.907); fine soil particles were significantly negatively correlated (−0.039) with soil total nitrogen (P < 0.01); clay soil particles were positively correlated (0.948*) with soil total nitrogen and soil organic carbon (0.932*); soil total nitrogen was significantly positively correlated (0.959**) with soil organic carbon. The comprehensive analysis shows that the correlation coefficient among the evaluation indexes was highest in poplar woodland, followed by natural grassland, lower in farmland and lemon woodland, and lowest in abandoned farmland.

Table 3 Correlation coefficient of soil physical and chemical properties indexes under land use types.

	 	X 1	X 2	X 3	X 4	X 5	X 6	X 7	X 8	X 9	
F	X 1	1									
X 2	–0.012	1								
X 3	–0.991**	0.021	1							
X 4	–0.380	–0.901*	0.343	1						
X 5	0.851	0.153	–0.905*	–0.379	1					
X 6	–0.819	–0.077	0.842	0.344	−0.757	1				
X 7	0.164	–0.114	–0.286	0.219	0.61	−0.203	1			
X 8	0.549	–0.441	–0.634	0.309	0.672	−0.716	0.679	1		
X 9	0.711	–0.193	–0.792	0.023	0.913*	−0.631	0.847		1	
Q	X 1	1									
X 2	0.433	1								
X 3	−0.833	−0.763	1							
X 4	−0.626	−0.531	0.414	1						
X 5	−0.578	0.184	0.211	0.348	1					
X 6	−0.142	0.501	−0.275	0.164	0.877	1				
X 7	0.371	0.351	−0.656	0.267	0.317	0.671	1			
X 8	0.265	−0.67	0.087	0.246	−0.741	−0.721	−0.05	1		
X 9	0.58	0.402	−0.811	0.15	−0.242	0.166	0.762	0.263	1	
G	X 1	1									
X 2	−0.668	1								
X 3	−0.893	0.313	1							
X 4	−0.329	−0.307	0.416	1						
X 5	−0.62	0.352	0.482	0.537	1					
X 6	0.477	−0.598	−0.364	0.323	−0.578	1				
X 7	0.627	−0.061	−0.68	−0.718	−0.902	0.417	1			
X 8	0.927*	−0.853	−0.75	0.027	−0.54	0.697	0.441	1		
X 9	0.702	−0.391	−0.58	−0.537	−0.993	0.598	0.92	0.617	1	
N	X 1	1									
X 2	−0.912	1								
X 3	−0.94	0.82	1							
X 4	−0.08	−0.296	0.01	1						
X 5	−0.207	0.365	0.257	−0.502	1					
X 6	0.548	−0.509	−0.241	−0.382	0.008	1				
X 7	0.144	0.148	−0.114	−0.728	0.895	0.167	1			
X 8	−0.508	0.784	0.307	−0.528	0.509	−0.576	0.515	1		
X 9	−0.164	0.215	−0.023	0.085	0.644	−0.622	0.531	0.521	1	
Y	X 1	1									
X 2	−0.258	1								
X 3	−0.942	−0.068	1							
X 4	−0.805	−0.236	0.871	1						
X 5	−0.333	0.606	0.083	0.246	1					
X 6	0.263	0.753	−0.483	−0.758	0.027	1				
X 7	0.108	−0.877	0.225	0.145	−0.75	−0.543	1			
X 8	−0.907	0.107	0.862	0.932*	0.392	−0.481	−0.164	1		
X 9	−0.942	−0.039	0.967	0.948*	0.208	−0.543	0.082	0.959**	1	
Notes.

F agricultural land

Q abandoned farmland land

G natural grassland

N artificial lemon woodland

Y poplar woodland

X1 coarse soil particles

X2 fine soil particles

X3 very fine soil particles

X4 clay soil particles

X5 soil water content

X6 soil pH

X7 soil electrical conductivity

X8 soil organic carbon

X9 total soil nitrogen

Comprehensive evaluation of soil quality

As shown in (Table 4), the first three principal components were extracted using the eigenvalue λ > 1 principle, with eigenvalues of 5, 2, and 1, respectively, and variance contribution rates of 55 percent, 18 percent, and 13 percent, respectively. In addition to satisfying the fundamental condition that the eigenvalue should be greater than 1 (λ > 1), the condition that the cumulative contribution of the first n principal components is at least 85 percent must also be satisfied. Only when both conditions are satisfied can the first n principal components be considered to have effectively reflected the information of the original variable. The cumulative contribution of the first three principal components in this study was 87 percent (>85 percent), implying that the first three principal components can represent all the information. The first principal component combined data from seven evaluation indicators with coefficients greater than 0.5, including fine soil particles (X2), very fine soil particles (X3), soil clay particles (X4), soil water content (X5), soil pH value (X6), soil electrical conductivity (X7), and soil total nitrogen (X9). The first principal component had the highest contribution rate and contained the most indicators, indicating that these soil indicators are critical for soil quality and can be used to explain 55 percent of the original nine soil quality factors. The second major component consisted of two indicators, soil coarse sand (X1) and very fine soil particles (X3), both of which had coefficients greater than 0.5 and accounted for 18 percent of the original soil’s quality. The third principal component consisted of two indicators: soil pH (X6) and soil organic carbon (X8), which together accounted for 13 percent of the original soil’s overall quality. The principal component score equation was established as follows using the principal component score matrix: (4) Z1=−0.009X1−0.171X2+0.136X3+0.171X4+0.177X5+0.135X6+0.178X7+0.099X8+0.176X9

(5) Z2=0.564X1−0.116X2−0.39X3−0.145X4+0.03X5+0.091X6−0.027X7+0.201X8+0.174X9

(6) Z3=0.027X1+0.196X2−0.169X3−0.207X4+0.299X5+0.47X6+0.281X7−0.58X8−0.095X9

where Z1, Z2, Z3 represent the three principal components, and X1−X9 represent the standardized variables of each evaluation index.

Table 4 Eigenvectors, eigenvalues, contribution rates and cumulative contribution rates of each factor in the principal component analysis.

 	1	2	3	4	5	6	7	8	9	
X 1	–0.043	0.964	0.032	–0.245	–0.067	–0.059	0.028	–0.004	0.000	
X 2	–0.858	–0.198	0.232	0.404	0.025	0.08	–0.017	0.009	0.000	
X 3	0.686	–0.667	–0.200	–0.085	0.099	–0.162	0.018	–0.040	0.000	
X 4	0.86	–0.248	–0.245	–0.305	–0.088	0.184	–0.041	0.048	0.000	
X 5	0.888	0.051	0.354	0.2	–0.129	–0.100	–0.044	0.125	0.000	
X 6	0.678	0.155	0.556	–0.076	0.444	0.048	–0.021	–0.016	0.000	
X 7	0.895	–0.046	0.332	0.145	–0.185	0.062	0.161	–0.045	0.000	
X 8	0.496	0.343	–0.686	0.325	0.232	0.021	0.064	0.044	0.000	
X 9	0.885	0.297	–0.113	0.272	–0.124	0.011	–0.127	–0.100	0.000	
Eigenvalues	5.027	1.708	1.183	0.569	0.34	0.087	0.052	0.034	0.000	
Contribution rate	55.858	18.982	13.15	6.317	3.78	0.964	0.572	0.376	0.000	
Cumulative contribution rate	55.858	74.841	87.99	94.307	98.087	99.051	99.624	100.000	100.000	
Notes.

X1 coarse soil particles

X2 fine soil particles

X3 very fine soil particles

X4 clay soil particles

X5 soil water content

X6 soil pH

X7 soil electrical conductivity

X8 soil organic carbon

X9 total soil nitrogen

The standardized variables were incorporated into these three functional equations to obtain the soil scores Z for various land-use types. The variance contribution rate of the factors was used as the weight for the comprehensive evaluation, and the soil quality score for each land-use type was calculated by weighing the respective variance contribution rate. The comprehensive score of soil quality was obtained using the formula; (7) F=0.559Z1+0.190Z2+0.132Z3.

As shown in (Table 5), farmland, abandoned farmland, natural grassland, lemon woodland, and poplar woodland all had comprehensive soil quality scores of 1.017, 0.102, −0.112, −0.336, and −0.671, respectively. Farmland had the highest comprehensive soil quality score, and only farmland and abandoned farmland had a positive comprehensive soil quality score, indicating that the soil quality of these two types of land use was above average, whereas the soil quality comprehensive scores of natural grassland, lemon woodland, and poplar forest land, were all negative and below average.

Table 5 Factor points of principal components and the comprehensive points.

Land use type	Z 1	Z 2	Z 3	F	Ranking	
F	1.725	–0.255	0.777	1.017	1	
Q	0.004	–0.675	0.102	0.102	2	
G	–0.507	0.334	–0.112	–0.112	3	
N	–0.642	–0.642	–1.312	–0.336	4	
Y	–1.147	–0.765	0.876	–0.671	5	
Notes.

F farmland

Q abandoned farmland

G natural grassland

N artificial lemon woodland

Y poplar woodland

Z1 first principal component factor score

Z2 second principal component factor score

Z3 third principal component factor score

F overall soil quality score

Discussion

The impact of land-use change on the physical and chemical properties of soil

Human activities and climate change have resulted in dramatic changes in land use, surface morphology, physical and chemical cycles, and ecological balance in recent years, resulting in significant changes in the global climate and irreversible species decline. Because land use has a significant impact on the environment’s sustainable development, it has become one of the primary drivers of global environmental change (Statuto, Cillis & Picuno, 2016), while also having a significant impact on the physical and chemical properties of soil to a degree.

The composition of soil particle size is critical in determining soil anti-erodibility (Chen et al., 2013), and various land-use practices reflect disturbance-induced changes in soil particle size distribution (Feng et al., 2016). In this study, the coarse soil content of the soil ranged from high to low as follows: natural grassland > abandoned farmland > farmland > lemon woodland > poplar woodland; the fine soil particles content of the soil ranged from high to low as follows: poplar woodland > natural grassland > abandoned farmland > lemon woodland > farmland, and the very fine sand content of the soil ranged from high to low as follows: farmland > lemon woodland > farmland. The primary reason is that the crops cultivated in the area is maize, which had grown to a certain height and was densely planted at the time of sampling, and the surface vegetation cover of abandoned farmland, natural grassland, and lemon woodland was relatively high, which can reduce wind speed near the surface, reduce wind erosion, fix clay soil particles on the surface, and block clay particles from being blown away by the wind. Farmland contained the most clayey soil particles, which may be due to the artificial application of organic fertilizers by farmers in the study area. However, due to drought and low rainfall in poplar forests, tree body growth is poor and the ground surface lacks effective coverage, resulting in severe wind erosion, which blows away soil particles and increases fine sand content.

Farmland had a higher soil water content, high pH value, and electrical conductivity than abandoned farmland, natural grassland, lemon woodland, and poplar woodland in this study. The reasons for this are as follows: the vegetation roots of abandoned farmland, natural grassland, lemon woodland, and poplar forest land fiercely compete for soil moisture, resulting in a lack of water in the topsoil and low water content, whereas farmland in the study area has been artificially irrigated for an extended period, resulting in a much higher surface water content (Zhang et al. 2019; Yao et al., 2020). Because the soil in this area has been salinized, the overall pH value is high and alkaline. Among them, abandoned farmland, natural grassland, and lemon woodland are less disturbed by humans, have a high cover of surface vegetation, low evaporation, which protects the soil from erosion, and low salinity, which results in a lower soil pH and electrical conductivity (Koetlisi & Muchaonyerwa, 2019).

Soil organic carbon and total nitrogen are important indicators of soil fertility because they are derived primarily from above- and below-ground plant litter decomposition and are heavily influenced by vegetation, climate, and anthropogenic activities (Fornara, Banin & Crawley, 2013; Yu et al., 2020). The order of the soil organic carbon content in this study was farmland > abandoned farmland > lemon woodland > farmland > natural grassland > Poplar woodland, and the order of the soil total nitrogen content was farmland > abandoned farmland > lemon woodland > natural Grassland > Poplar forest land. The soil organic carbon and total nitrogen contents of poplar forest land were the lowest because the majority of poplar forest land in this area has been planted as farmland protection forests, and the drought and low rainfall have caused them to grow poorly, with some dying, leaving the top layer soil highly susceptible to erosion and degradation. Farmland had a higher total nitrogen content than abandoned farmland, grassland, lemon woodland, and poplar woodland because farmland in this area is carefully managed to restore the soil’s original fertility through fertilization, irrigation, and crop growth. However, tilling and reclamation of farmland’s surface soil promotes soil respiration and accelerates carbon element decomposition, or because a significant amount of above-ground or underground farmland is removed during crop harvesting, and the proportion of carbon elements fixed by primary production allocated to the soil is low, reducing the organic carbon content (Kumar et al., 2017). The higher soil organic carbon and total nitrogen contents of abandoned farmland, natural grassland, and lemon woodland are a result of their high vegetation cover, which effectively prevents erosion and promotes organic matter accumulation (Batjes, 2014; Gerschlauer et al., 2019).

Correlation between soil quality indicators under conditions of land-use change

The interdependence of land quality indicators varies according to land-use type. The interaction and coordination effects between soil quality indicators can provide a comprehensive picture of the soil’s productivity and adaptability to adversity (Paz-Ferreiro & Fu, 2016).

To begin with, the composition of soil particle size has a significant effect on the structure and properties of the soil, affecting water absorption, anion and cation exchange, and the supply of nutrient elements such as carbon and nitrogen. There was a negative correlation between very fine soil particles and soil moisture content in farmland in this study, which is likely due to the higher content of very fine particles in farmland compared to abandoned farmland, natural grassland, lemon woodland, and poplar forest land, all of which increase water infiltration and decrease surface water content (Yost & Hartemink, 2019). There was a significantly negative correlation between soil coarse particles, soil organic carbon, and soil total nitrogen in poplar forest land, but a significantly positive correlation between very fine soil particles, soil clay particles, and soil organic carbon in poplar forest land. This is because clay particles in soil are the primary inorganic colloids, are cemented with organic matter, and provide a cementing environment as well as potentially acting as a protective factor for the soil’s structural properties (Latifi et al., 2018). However, there was a positive correlation between soil coarse particles and soil total nitrogen in the natural grassland, which may be explained by the error introduced by fewer experimental replications in this study.

Secondly, the amount of water in the soil has a direct effect on the dissolution of various salts, material transformation, and decomposition of organic matter in the soil (Lu et al., 2011; Fornara, Tilman & Houlton, 2012). The soil water content and electrical conductivity of lemon woodland were positively correlated in this study, owing to the complete dissolution of various salts in the soil, whereas the soil water content and electrical conductivity of natural grassland, farmland, abandoned farmland, and poplar forest land was negatively correlated or even uncorrelated. Additionally, the negative correlation between soil water content and soil organic carbon content in natural grassland is because soil salinization impairs soil nutrient cycling, resulting in a decline in soil fertility and thus a decrease in soil organic carbon content (Herbert et al., 2015). By contrast, the soil water content of farmland was high due to anthropogenic irrigation, and the soil’s relatively low salt content, which does not correlate with soil electrical conductivity. However, a certain amount of soil water facilitates the preservation of soil structure, which increases soil sorption, and increases nutrient uptake and soil solidity. The soil’s adsorption capacity increases nutrient absorption and stability, thereby increasing the soil’s organic carbon content (Amer, 2019).

Thirdly, a positive correlation existed between soil electrical conductivity and total soil nitrogen, which could be explained by an increase in soil electrical conductivity and an increase in the rate of organic matter accumulation and mineralization by the soil (Ding et al., 2020). This correlation was strongest in natural grasslands, but not in farmland, lemon woodlands, or poplar woodlands, most likely because natural grasslands have more surface vegetation and deadfalls, which are a major source of organic matter and total nitrogen in the soil.

Fourthly, soil organic carbon and total nitrogen content are important indicators of soil fertility and productivity, and changes in soil organic carbon and total nitrogen content are strongly influenced by land use and soil management (Negasa, 2020; Zhang et al., 2020). While there was a significant positive correlation between soil organic carbon and soil total nitrogen in poplar forest land, no correlation was observed in the other sample plots. The reason for this may be due to the effect of various land-use management factors on nitrogen mineralization in soil, clay mineral fixation, nitrification, and denitrification processes, as well as leaching via gaseous and post-solubilization, nitrogen fixation by plants, and competitive nitrogen utilization by plants and microorganisms. On the one hand, it demonstrates that soil organic carbon and total nitrogen enrichment in farmland, abandoned farmland, natural grassland, and lemon woodland are dynamic processes, as topsoil organic carbon and total nitrogen are more susceptible to disturbance by factors such as stand type, climatic environment, apoplankton abundance, and degree of decomposition (Zhang et al., 2021). For example, tillage activities, abandoned farmland and natural grassland restoration processes, as well as the evolution of lemon woodland, all have a profound effect on the relationship between soil organic carbon and total nitrogen in the soil. Simultaneously, the sparse herbaceous vegetation covers in the understory of the poplar woodland, the low levels of soil organic carbon and total nitrogen, and the low number and relatively low activity of soil microorganisms maintain a relative equilibrium between soil organic carbon and total nitrogen, and when the relative equilibrium is reached, the soil nitrogen content is lagged.

Finally, due to the study area’s aridity and degraded ecological environment, woody plants in poplar forests and lemon woodland require a high amount of water and nutrients from the soil, which results in a high correlation between its various soil quality indicators and relatively fragile soil quality. While natural grasslands and farmlands experience a high level of anthropogenic disturbances such as irrigation, animal husbandry, and so on, which has a greater impact on the relationship between their soil quality indicators, abandoned lands experience fewer human-induced and natural disturbances and have sparse surface vegetation, resulting in a low correlation between soil quality indicators.

The impact of land-use changes on soil quality

The purpose of soil quality evaluation is to gain a thorough understanding of the soil and to reflect changes in soil management to manage and protect it effectively (Rachman et al., 2020). As can be seen from the above results, the soil quality comprehensive scores for the five different land-use types in YongDeng County are ranked as follows: farmland > abandoned farmland > 0 > natural grassland > lemon woodland > poplar woodland, where a higher comprehensive score indicates a higher degree of soil quality under that land use type, and vice versa. The composite scores of farmland and abandoned farmland were positive, indicating that the soil quality under these vegetation types is higher than the average level, while the composite scores of grassland, lemon woodland, and poplar woodland were negative, indicating below-average soil quality.

This result is because the farmland in this area has a certain soil fertilization effect as a result of a series of fine management measures such as rational fertilization and irrigation, which effectively improves soil fertility and prevents soil degradation, and thus the farmland had the highest soil quality score and was higher than average. Secondly, natural grassland had a lower soil quality score due to the influence of soil salinization in the area as this distresses the soil respiration, nitrogen cycle, and decomposing functionality of soil microorganisms (Singh, 2016). Furthermore, it affects the vegetation growth directly by reducing the grass plant water uptake (osmotic stress) and/or by deteriorating the transpiring leaves (specific ion effects) (Parihar et al., 2015), in turn reducing organic input to the soil. However, naturally restored grassland retains some fertility after abandonment, which can help improve soil fertility to some extent (Deru et al., 2018; Li et al., 2021). Finally, compared to grassland, lemon woodland and poplar woodlands extract more nutrients from the soil, lowering soil quality. However, because of the obvious fertility island effects, the soil quality of lemon woodland was higher than that of poplar woodland.

As an ecological transition zone separating agricultural areas in eastern China from grassland-pastoral areas in the west, YongDeng County is a marginal area for traditional cultivation due to its harsh natural conditions, low annual rainfall, and high inter-annual variability. To enhance agricultural and pastoral productivity and ecological conditions, the area’s land-use structure should be adjusted. The geographical environment of the sample plots in the study area is quite different, but the principle of adapting measures to local conditions should be followed.

Conclusions

The effect of various land-use changes on the physicochemical properties and quality of soil in YongDeng County is evaluated in this study. The study revealed that land-use change has a significant effect on the quality and spatial distribution of soil physicochemical properties. In the study area, the soil quality of farmland that has been finely managed and naturally restored to grassland following abandonment is superior, whereas the soil quality of natural grassland, artificial lemon woodland, and poplar forest land is substandard. The comprehensive analysis demonstrates that conservation tillage and fine management of water-irrigated farmland, as well as the natural conversion of abandoned farmland to grassland, can improve the soil quality of sandy soils, reduce water and soil loss, increase fertility, and gradually improve regional ecological environmental conditions.

Supplemental Information

Supplemental Information 1 Raw measured values from laboratory analysis and field

Click here for additional data file.

Special thanks to the College of Forestry and College of Resources and Environment, Gansu Agricultural University for all the support in providing a laboratory space for the conduct of experiments.

Additional Information and Declarations

Competing Interests

Author Contributions

Data Availability

The authors declare that they have no known competing financial interests or personal relationships that could have appeared to influence the work reported in this paper.

Samuel Adingo and Jie-Ru Yu conceived and designed the experiments, performed the experiments, analyzed the data, prepared figures and/or tables, authored or reviewed drafts of the paper, and approved the final draft.

Liu Xuelu and Zhang Xiaoning conceived and designed the experiments, performed the experiments, prepared figures and/or tables, and approved the final draft.

Sun Jing conceived and designed the experiments, analyzed the data, authored or reviewed drafts of the paper, and approved the final draft.

Xiaodan Li conceived and designed the experiments, analyzed the data, prepared figures and/or tables, authored or reviewed drafts of the paper, and approved the final draft.

The following information was supplied regarding data availability:

The raw values of measurement are available in the Supplementary File.

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
