# Peer review of "Land-use change influence soil quality parameters at an ecologically fragile area of YongDeng County of Gansu Province, China"

_PeerJ, doi:10.7717/peerj.12246_

## Round 0.1 · original submission · Minor Revisions

Dear Dr. Adingo,

Thank you for your submission to PeerJ.

It is my opinion as the Academic Editor for your article - Land-use change influence soil quality parameters at an ecologically fragile area: A case study of YongDeng County of Gansu Province, China. - that it requires a number of Minor Revisions.

The manuscript has been assessed by three reviewers. Two of the three reviewers agree on the fact that there are a few points that need to be addressed. I agree with this evaluation and I would, therefore, request for the manuscript to be revised accordingly.

My suggested changes and reviewer comments are shown below and on your article 'Overview' screen.In addition, one of the reviewers has attached an annotated manuscript to their review.

Please address these changes and resubmit. Although not a hard deadline please try to submit your revision within the next 21 days.

With kind regards,
Chenxi Li
Academic Editor, PeerJ

Reviewer 1 ·

Basic reporting

no comment

Experimental design

Experimental design; rewrite it to make it easy to follow, you can use illustration of figure to present your experimental design

Validity of the findings

The abstract is well written and clear to the reader, congratulation for good job.
Introduction: Well structured and well written with enough background information

Additional comments

1. Abstract: The abstract is well written and clear to the reader, congratulation for good job.
2. Introduction: Well structured and well written with enough background information
3. Materials and Methods
4. Line 111 to 112; present degrees Celsius by using symbol
5. Line 130: the area is 15.3hm2, it is not clear
6. Line 130 and other places in the Ms i.e., plants spaced 7m leave space between value and unit
7. Line 135; Table 1 summarizes the fundamental characteristics of each land-use type, you presented the table caption but the table itself is not located in a particular location. I understand figures and tables are at the end of the document, thus in the text just cite the figure or table without full figure or table caption.
8. Line 137 to 147: Experimental design; rewrite it to make it easy to follow, you can use illustration of figure to present your experimental design.
9. Line 151; How did you determined the soil moisture? Did you use soil core to take the soil samples for moisture content?
10. Line 216: why you decided to use spearman’s correlation? and not Pearson?
11. What is the recommended SOC ad Total N in the area for agricultural productivity?
12. Conclusion: acceptable

·

Basic reporting

No comments

Experimental design

No comments

Validity of the findings

No comments

Additional comments

This paper has high standard interms of original research, experiments conducted and results presented by the authors. This paper addresses issues in dry land, abandoned land and other types that are critical for food sustainability.

Reviewer 3 ·

Basic reporting

I thank the authors for their extensive dataset (figures and tables) as well as the raw data that they helped to track the statistical analysis collected from detailed field studies. The articles are written in English at a high level, In addition, the manuscript is clearly written professionally, but requires revision. Such as references for literature that need to be finalized.
Hypotheses is consistent with the results obtained. All notes are given in the text of the article.

Experimental design

The goals and objectives are reflected in this work within the goals of the journal. The research question is clear, relevant and meaningful. Shows how research fills the identified knowledge gap.
The methods are described with sufficient detail and information to reproduce, but there are minor issues with the use of methods for thinning such as soil texture. See comments in the pdf file.

Validity of the findings

All basic data were provided; they are reliable, statistically reliable and controlled.
Conclusions are well articulated, related to the original research question, and limited to supporting results.

Additional comments

The structure of the article is written at a high level, but needs improvement. All remarks are given in the text of the article. Determining the type of soil texture is of particular concern to scientists when talking about the quality of soil parameters.

I commend the authors for their extensive data set and also the raw data they helped track the statistical analysis, compiled over of detailed fieldwork. In addition, the manuscript is clearly written in professional. If there is a weakness, it is in the methodology (as I have noted in text) which should be improved upon before acceptance.

Annotated reviews are not available for download in order to protect the identity of reviewers who chose to remain anonymous.

---

## Round 0.2 · accepted · Accept

Dear Dr. Adingo,

Thank you for your submission to PeerJ.

I am writing to inform you that your manuscript - Land-use change influence soil quality parameters at an ecologically fragile area of YongDeng County of Gansu Province, China - has been Accepted for publication. Congratulations!

The authors addressed the reviewers' concerns and substantially improved the content of the manuscript.

So, based on my own assessment as an editor, no further revisions are required and the manuscript can be accepted in its current form.

This is an editorial acceptance; publication is dependent on authors meeting all journal policies and guidelines.

Next steps: Your article is being checked and you will receive a list of production tasks shortly. After you complete these tasks, your proofing PDF will be created (please do not proof check your reviewing PDF!).
This is great news, why not share it with a tweet!

Congratulations again, and thank you for your submission.
With kind regards,
Chenxi Li
Academic Editor, PeerJ

Reviewer 1 ·

Basic reporting

Good

Experimental design

well presented now

Validity of the findings

Acceptable

Additional comments

The authors have followed and worked on all comments and suggestions provided and the manuscript quality and clarity are impressively improved. ACCEPT FOR PUBLICATION

Reviewer 3 ·

Basic reporting

I thank the authors for their extensive dataset (figures and tables) as well as the raw data that they helped to track the statistical analysis collected from detailed field studies. The articles are written in English at a high level.

Experimental design

The goals and objectives are reflected in this work within the goals of the journal. The research question is clear, relevant and meaningful. Shows how research fills the identified knowledge gap.
The methods are described with sufficient detail and information to reproduce.

Validity of the findings

All basic data were provided; they are reliable, statistically reliable and controlled.
Conclusions are well articulated, related to the original research question, and limited to supporting results.

Additional comments

The structure of the article is written at a high level.